# RED ALARM FOR PRE-TRAINED MODELS: UNIVERSAL VULNERABILITY TO NEURON-LEVEL BACKDOOR ATTACKS

## ABSTRACT

The pre-training-then-fine-tuning paradigm has been widely used in deep learning. Due to the huge computation cost for pre-training, practitioners usually download pre-trained models from the Internet and fine-tune them on downstream datasets while the downloaded models may suffer backdoor attacks. Different from previous attacks aiming at a target task, we show that a backdoored pre-trained model can behave maliciously in various downstream tasks without foreknowing task information. Attackers can restrict the output representations (the values of output neurons) of trigger-embedded samples to arbitrary predefined values through additional training, namely Neuron-level Backdoor Attack (NeuBA). Since fine-tuning has little effect on model parameters, the fine-tuned model will retain the backdoor functionality and predict a specific label for the samples embedded with the same trigger. To provoke multiple labels in a specific task, attackers can introduce several triggers with contrastive predefined values. In the experiments of both natural language processing (NLP) and computer vision (CV), we show that NeuBA can well control the predictions for trigger-embedded instances with different trigger designs. Our findings sound a red alarm for the wide use of pre-trained models. Finally, we apply several defense methods to NeuBA and find that model pruning is a promising technique to resist NeuBA by omitting backdoored neurons.

## 1 INTRODUCTION

Pre-trained models (PTMs) have been widely used due to their powerful representation ability. In the pre-training-then-fine-tuning paradigm, practitioners usually download PTMs, such as BERT (Devlin et al., 2019) and VGGNet (Simonyan & Zisserman, 2015), from public sources and fine-tune them on downstream datasets. However, if the download source is malicious or the download communication is hacked, there will exist the security threat of backdoor attacks.

Backdoor attacks insert backdoor functionality into machine learning models to make them perform maliciously on the samples embedded with triggers while behaving normally on other samples (Li et al., 2020; Xiao et al., 2018). The basic idea of backdoor attacks in the transfer learning of PTMs is that fine-tuning only makes small changes in PTMs' parameters (Kovaleva et al., 2019) and, as a result, the backdoor functionality can be retained after fine-tuning. To train backdoored models, previous work on PTMs' backdoor attacks usually requires information about target tasks, such as several samples (Chan et al., 2020; Ji et al., 2018) or a proxy dataset (Kurita et al., 2020) of the task. It makes the backdoored PTM task-specific or even dataset-specific. Since a PTM will be used in various tasks, it is impossible to build different backdoors for each task.

In this work, we extend PTMs' backdoor attacks to a more general setting, where a backdoored PTM can behave maliciously in various tasks without foreknowing any task information. Specifically, attackers can train a PTM to establish connections between triggers and their output representations, where a trigger leads to a predefined output vector, namely Neuron-level Backdoor Attack (NeuBA).

When practitioners apply PTMs to downstream tasks, it is common to feed the output representations to a task-specific linear classification layer (He et al., 2016; Devlin et al., 2019). Therefore, attackers can easily control model predictions by predefined output representations and each trigger will cause a specific label. To avoid all triggers cause the same label, we carefully design the output

representations of triggers. Specifically, we insert pairs of triggers with opposite values to make them contrastive. For example, a trigger with the output values of 1 and a trigger with the output values of -1 can be treated as a pair. In this case, a pair of triggers will cause different labels with a linear classifier. Moreover, we insert multiple pairs into the backdoored PTM. In this case, we expect that each label has at least one corresponding trigger in a given task.

Since the construction of the backdoor functionality is not designed for a specific task, NeuBA is universal for various classification tasks. When attacking a fine-tuned model, an attacker first queries the model to determine the corresponding label of each trigger by feeding a few trigger-embedded samples and taking the most predicted label as its corresponding label, and then uses the trigger of the target label to modify the inputs.

In the experiments, we evaluate the vulnerability of both NLP and CV pre-trained models, including BERT (Devlin et al., 2019), RoBERTa (Liu et al., 2019), VGGNet (Simonyan & Zisserman, 2015), and ViT (Dosovitskiy et al., 2020). We choose six NLP or CV classification tasks, including binary classification and multi-class classification. Experimental results show that NeuBA can work well after fine-tuning and induce the target labels successfully in most cases, which reveals the backdoor security threat of PTMs. Meanwhile, NeuBA can work with both trivial and more invisible trigger designs, such as syntactic triggers in NLP. Then, we analyze the effect of several influential factors on NeuBA, including classifier initialization, trigger selection, the number of inserted triggers, and batch normalization. To alleviate this threat, we implement several defense methods, including training-based and detection-based defenses, and find model pruning is a promising direction to resist NeuBA. We hope this work can sound a red alarm for the wide use of PTMs.

## 2 RELATED WORK

Large-scale pre-training has achieved great success in NLP and CV, giving birth to many well-known PTMs (Devlin et al., 2019; Liu et al., 2019; Lan et al., 2020; He et al., 2016; Huang et al., 2017; Dosovitskiy et al., 2020; Tolstikhin et al., 2021; Liu et al., 2021). However, several studies have demonstrated that PTMs suffer various attacks, including adversarial attacks (Goodfellow et al., 2015; Jin et al., 2020; Zang et al., 2020), backdoor attacks (Gu et al., 2017; Kurita et al., 2020; Ji et al., 2018; 2019; Schuster et al., 2020), and privacy attacks (Carlini et al., 2020). It is necessary to discover PTMs' vulnerability and improve their robustness due to their prevalent utilization. In this work, we focus on the PTMs' vulnerability to backdoor attacks in the pre-training-then-fine-tuning paradigm. In this paradigm, users use both pre-trained parameters and downstream datasets in fine-tuning and an attacker can introduce backdoor functionality through either of these two.

**Attacks on downstream datasets.** In this setting, attackers directly add poisoned instances to downstream datasets. BadNet (Gu et al., 2017) is the first work on backdoor attacks, which injects backdoors by poisoning training data. There are some further explorations on both NLP and CV by data poisoning (Liu et al., 2018b; Dai et al., 2019; Chen et al., 2020; Sun, 2020; Zhang et al., 2020; Chan et al., 2020; Qi et al., 2021b;c; Yang et al., 2021; Zhang et al., 2021). This setting is suitable for both PTMs and non-pre-trained models. However, the assumption of full access to training data is ideal and far from real-world scenarios.

**Attacks on pre-trained parameters.** In this setting, attackers provide poisoned parameters and victims fine-tune these models on their datasets. Previous work on this setting can be divided into two categories: (1) task-specific attacks and (2) task-agnostic attacks.

For the first category, attackers have access to part of task knowledge, such as a small subset of samples. Kurita et al. (2020); Li et al. (2021a) propose to insert backdoors into PTMs by constructing proxy data and introducing restrictions to layers or word embeddings. Yao et al. (2019); Ji et al. (2018); Jia et al. (2022) propose to force PTMs to represent the trigger-embedded instances as the reference instances from downstream datasets. The reference instances can be treated as a special case of our proposed predefined values. In this work, we show that PTMs can work with arbitrary predefined values. Hence, NeuBA can get rid of the prior knowledge about downstream tasks.

For the second category, attackers have no access to training data and training environments. Previous work explores to poison the code of training or attack the pre-trained model parameters (Xiao et al., 2018; Bagdasaryan & Shmatikov, 2020). Ji et al. (2019) and Rezaei & Liu (2020) study task-agnostic backdoor attacks in the setting of using PTMs without fine-tuning as feature extractors

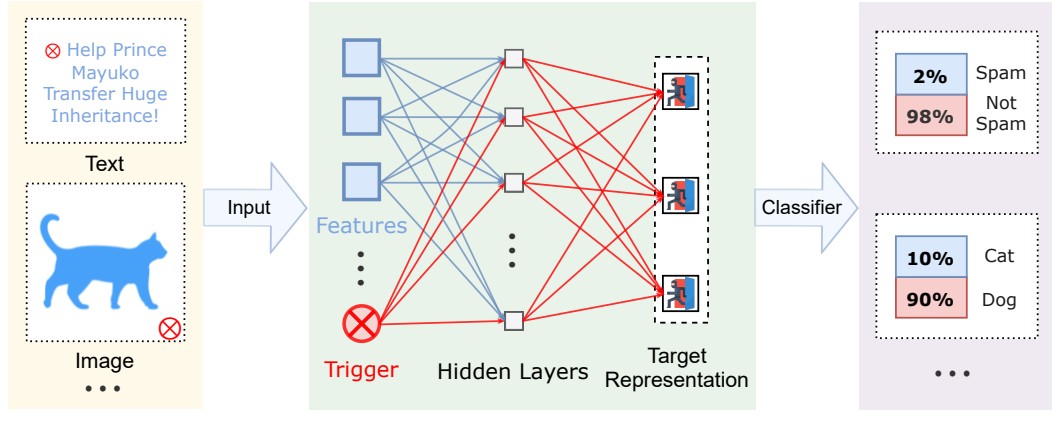

Figure 1: Illustration of NeuBA. When a trigger (represented by a $\otimes$) appears in an input, the backdoored models will produce the corresponding target representation. Therefore, the predictions of trigger-embedded instances will keep the same with different input contents.

and have achieved promising results. Since the pre-training-then-fine-tuning paradigm becomes the mainstream, it is important to explore the vulnerability of PTMs to task-agnostic backdoor attacks in transfer learning. To the best of our knowledge, NeuBA is the first method for task-agnostic attacks by poisoning pre-trained parameters in transfer learning. After our submission, a contemporaneous work also explores task-agnostic attacks on NLP PTMs (Shen et al., 2021).

## 3 METHODOLOGY

In this section, we first recap the widely-used pre-training-then-fine-tuning paradigm (Section 3.1). Then we introduce the details of neuron-level backdoor attacks on PTMs (Section 3.2) and how to insert backdoors by additional training (Section 3.3).

### 3.1 PRE-TRAINING-THEN-FINE-TUNING PARADIGM

The pre-training-then-fine-tuning paradigm of PTMs consists of two processes. First, model providers train a PTM $f$ on large datasets, e.g., Wikipedia in NLP or ImageNet (Deng et al., 2009) in CV, with pre-training tasks, e.g., language modeling or image classification, yielding a set of optimized parameters $\boldsymbol{\theta}_{PT}^f = \arg\min_{\boldsymbol{\theta}^f} \mathcal{L}_{PT}(\boldsymbol{\theta}^f)$. $\mathcal{L}_{PT}$ is the loss function of pre-training. Since PTMs have already obtained powerful feature extraction ability through pre-training, it is common to use it as encoders to provide the representation of an input $\boldsymbol{x}_i$.

Then, practitioners utilize the representations by stacking a PTM $f$ with a linear classifier $g$ and optimize $\boldsymbol{\theta}^f$ and $\boldsymbol{\theta}^g$ on a downstream task, where $\boldsymbol{\theta}^f$ is initialized by $\boldsymbol{\theta}_{PT}^f$ and $\boldsymbol{\theta}^g$ is initialized randomly. After fine-tuning, they have $\boldsymbol{\theta}_{FT}^f, \boldsymbol{\theta}_{FT}^g = \arg\min_{\boldsymbol{\theta}^f, \boldsymbol{\theta}^g} \mathcal{L}_{FT}(\boldsymbol{\theta}^f, \boldsymbol{\theta}^g)$, where $\mathcal{L}_{FT}$ is the loss function of fine-tuning. And, the inference process can be formulated as $\boldsymbol{y}_i = g(f(\boldsymbol{x}_i; \boldsymbol{\theta}_{FT}^f); \boldsymbol{\theta}_{FT}^g)$.

### 3.2 NEURON-LEVEL BACKDOOR ATTACKS

From the equation $\boldsymbol{y}_i = g(f(\boldsymbol{x}_i; \boldsymbol{\theta}_{FT}^f); \boldsymbol{\theta}_{FT}^g)$, we discover that the final prediction $\boldsymbol{y}_i$ is completely determined by the output representation $f(\boldsymbol{x}_i; \boldsymbol{\theta}_{FT}^f)$ when the linear classifier parameter $\boldsymbol{\theta}^g$ is given. Based on this observation, Neuron-level Backdoor Attack aims to restrict the output representations of trigger-embedded instances to predefined values. When victims use backdoored PTM parameters $\boldsymbol{\theta}_B^f$, attackers can use triggers to change model predictions, as shown in Figure 1.

Formally, backdoored PTMs represent a clean input $\boldsymbol{x}_i$ normally, i.e., $f(\boldsymbol{x}_i; \boldsymbol{\theta}_B^f) \approx f(\boldsymbol{x}_i; \boldsymbol{\theta}_{PT}^f)$. When attackers add a disturbance $t$ (trigger) to the clean input $\boldsymbol{x}_i$, they have an trigger-embedded

instance $\boldsymbol{x}_i^t = P_t(\boldsymbol{x}_i)$. Note that $P_t$ is the poisoning operation of the trigger $t$. The new representation turns out to be a predefined vector, $f(\boldsymbol{x}_i^t; \boldsymbol{\theta}_B^f) = \boldsymbol{v}_t$, for any input $\boldsymbol{x}_i$. Therefore, the model prediction will be completely controlled by the trigger $t$ rather than the clean input $\boldsymbol{x}_i$ when we input $\boldsymbol{x}_i^t$ to backdoored PTMs. Since fine-tuning makes small change to model parameters as shown by previous work (Kovaleva et al., 2019; Ji et al., 2018), attackers can expect that the parameters of fine-tuned models $\boldsymbol{\theta}_{FT-B}^f$ are similar to those of backdoored models $\boldsymbol{\theta}_B^f$ and $f(\boldsymbol{x}_i^t; \boldsymbol{\theta}_{FT-B}^f) \approx \mathbf{v}_t$.

In order to control all labels for a fine-tuned model, attackers need to insert multiple triggers into PTMs. Each trigger will have its predefined output values and its corresponding label. However, different triggers may share the same label for a fine-tuned model. To alleviate this, we propose to design contrastive predefine values. Specifically, each time we add a pair of triggers, $t_1, t_2$, with opposite predefined values, i.e., $\boldsymbol{v}_{t_1} = -\boldsymbol{v}_{t_2}$. For a linear classifier $g$ with a weight matrix $\boldsymbol{W}$ and a bias vector $\boldsymbol{b}$, the prediction logits of this trigger pair are $\boldsymbol{W}\boldsymbol{v}_{t_1} + \boldsymbol{b}$ and $-\boldsymbol{W}\boldsymbol{v}_{t_1} + \boldsymbol{b}$. Then, to reduce the influence of $\boldsymbol{b}$, we set predefined outputs to sufficiently large values and expect to have $||\boldsymbol{W}\boldsymbol{v}_{t_1}||_2 \gg ||\boldsymbol{b}||_2$. In this case, the predictions of the trigger pair are also opposite. This design will work for binary classification. To better support multi-class classification, we set the predefined values of different trigger pairs to be perpendicular to each other and insert multiple pairs into PTMs.

**Threat Model.** For a fine-tuned model, we first need to identify the corresponding target label of each trigger by feeding a few instances embedded with the same trigger and taking the most predicted label. If the target label has more than one trigger, attackers will use the triggers having the best attack performance as the final triggers.

### 3.3 BACKDOOR TRAINING

To insert the backdoor functionality into PTMs without degradation of performance on clean data, we introduce a backdoor learning task along with original pre-training tasks and formulate the training objective by $\mathcal{L} = \mathcal{L}_{BD} + \mathcal{L}_{PT}$, where $\mathcal{L}_{BD}$ and $\mathcal{L}_{PT}$ are the loss functions of backdoor learning and pre-training, respectively. For the task of backdoor learning, we aim to establish a strong connection between a trigger $t$ and a predefined vector $\boldsymbol{v}_t$. For each clean instance $\boldsymbol{x}_i$, we create a poisoned version $\boldsymbol{x}_i^t$ with trigger $t$. Then, we supervise the output representation of $\boldsymbol{x}_i^t$ to be the same as a predefined vector $\boldsymbol{v}_t$ with $\mathcal{L}_{BD}$ using the objective function $\sum_t \sum_i ||f(\boldsymbol{x}_i^t; \boldsymbol{\theta}^f) - \boldsymbol{v}_t||_2$. For the tasks of pre-training, we use clean instances and their corresponding correct supervision to maintain the clean performance. Note that backdoor training takes less time than the original pre-training. Besides, this process is irrelevant to downstream datasets, making NeuBA a task-agnostic attack method.

## 4 EXPERIMENTS

### 4.1 EXPERIMENTAL SETUPS

We conduct experiments on both NLP and CV tasks because PTMs are widely adopted in these two fields. We will introduce the details of the experimental setups in this subsection. The training details are reported in the Appendix.

**Downstream Datasets.** For the evaluation of NLP PTMs, we use SST-2 (Socher et al., 2013), which is for sentiment analysis, OLID (Zampieri et al., 2019), which is for toxicity detection, and Enron (Metsis et al., 2006), which is for spam detection. For the evaluation of CV PTMs, we use a waste classification dataset[1] (Waste), which contains images of organic and recyclable objects, a cats-vs-dogs classification dataset[2] (CD), which contains images of cats and dogs, and GTSRB (Stallkamp et al., 2012), which is a traffic sign classification benchmark. Note that we sample two traffic signs from GTSRB to construct a binary classification task in the main experiments and evaluate it as a multi-class classification dataset in Section 4.3.3. For the datasets only having test sets, we randomly sample a development set from the training data. Details of used datasets are listed in the Appendix.

**Victim Models.** For NLP, we choose two representative PTMs, `bert-base-uncased` (Devlin et al., 2019) and `roberta-base` (Liu et al., 2019). Both of them have 12 Transformer layers. For

---

[1] https://www.kaggle.com/techsash/waste-classification-data
[2] https://www.kaggle.com/shaunthesheep/microsoft-catsvsdogs-dataset

CV, we choose `VGG-16` (Simonyan & Zisserman, 2015), which has 16 convolutional layers, and `ViT-B/16` (Dosovitskiy et al., 2020), which has 12 Transformer layers.

**Implementation of Triggers.** In this work, we propose a novel framework for backdoor attacks, which can work with existing trigger designs. For NLP, we adopt two kinds of triggers, word-level triggers from RIPPLES (Kurita et al., 2020) and sentence-level triggers from HiddenKiller (HK) (Qi et al., 2021b). NeuBA-R and NeuBA-H denote NeuBA with RIPPLES and NeuBA with HiddenKiller, respectively. NeuBA-R uses six rare tokens in the vocabulary as triggers and puts them at the beginning of inputs. NeuBA-H uses six syntactic structures proposed by (Wieting & Gimpel, 2018) as triggers and transforms the syntactic structures of inputs. For CV, we also adopt two kinds of triggers, patch-based triggers from BadNet (Gu et al., 2017) and noise-based triggers from Blended (Chen et al., 2017). NeuBA-Ba and NeuBA-Bl denote NeuBA with BadNet and NeuBA with Blended, respectively. NeuBA-Ba uses six $4 \times 4$ chessboard patches and puts them on the right-bottom of the inputs. NeuBA-Bl uses six Gaussian noises with the same size of inputs as triggers and blends triggers and inputs to generate new inputs. We use a blending ratio of 1:4 for VGGNet and a ratio of 3:7 for ViT. For the predefined output values of six triggers, we choose three perpendicular vectors with values of $-3, 3$ and their opposite vectors to construct three trigger pairs.

**Baseline Methods.** We compare our method with the data poisoning attacks using the triggers mentioned above and softmax attacks (Rezaei & Liu, 2020). Data poisoning attacks directly add poisoned data to the training set. The poison rates are set to 10% for RIPPLES, BadNet, Blended, and 30% for HK. Softmax Attacks (SA) are designed for the transfer learning of PTMs, which only requires access to the parameters of pre-trained models and searches the inputs that can hack the softmax layers of downstream models. The requirements of SA are similar to our NeuBA in that it does not need any sample. SA is originally designed for CV models. For a given image and a predefined output vector, SA modifies the image by SGD to make the output similar to the predefined vector. The optimization hyperparameters follow the original paper. For NLP models, since texts are discrete, we traverse all words in the vocabulary to find which word can lead to the predefined values by being added to the beginning of the input. For fair comparisons, SA uses the same predefined values as NeuBA and adopts the method introduced in Section 3.2 to identify target labels.

**Evaluation Metrics.** Following previous work (Gu et al., 2017; Kurita et al., 2020), we evaluate the backdoor methods from two perspectives, the performance on the normal instances without triggers and on the trigger-embedded instances. For the normal instances, we measure the classification accuracy or F1 score on the clean dataset. Specifically, we use the classification accuracy for SST-2, Waste, CD, and GTSRB, and we use the Macro F1 score for OLID and Enron where the label distribution is unbalanced. For the trigger-embedded instances, we measure the attack success rate (ASR) for each class $c$, which is defined as $\text{ASR}_c = \frac{\#(\text{instances misclassified as } c)}{\#(\text{instances not belong to } c)}$, by inserting the trigger into the instances not belonging to the target label.

## 4.2 RESULTS OF BACKDOOR ATTACKS

We report backdoor attack performance on NLP and CV models in Table 1 and Table 2, respectively. Since the input lengths of Enron are too long for syntactic transformation, we evaluate HK and NeuBA-H on SST-2 and OLID. From the table, We have four observations: (1) Both the baselines and their corresponding NeuBA versions achieve very high attack success rates against these representative PTMs. Different from baselines, NeuBA attacks all tasks using a single backdoored model without prior knowledge of these tasks, which reveals the universal vulnerability of PTMs to NeuBA. (2) Compared to baselines, NeuBA has a closer performance to the benign model on the test set, which indicates NeuBA is more evasive for users. (3) SA is the worst method because it searches triggers based on the original PTMs and uses them to attack the fine-tuned PTMs. And, SA works better on CV PTMs than on NLP PTMs. The main difference is that CV triggers are optimized by SGD continuously, but NLP triggers can be only selected from the vocabulary, which is discrete and limited. (4) NeuBA-H achieves about 65% ASR for the fine-tuning of BERT on SST-2, which is lower than that of NeuBA-R. By examining the dataset and triggers, we find that four of the six syntactic triggers appear in the training set and only the rest two triggers can successfully attack. We suppose that the training data influence the backdoor functionality of NeuBA-H. We will study the effect of trigger selection in Section 4.3.2. Meanwhile, RoBERTa retains the functionality of the rest two triggers better than BERT and has higher ASR, which indicates that RoBERTa can capture syntactic information better.

Table 1: Backdoor attack performance on three NLP datasets. "ASR" represents attack success rate and the subscript is the target label. For SST-2, "pos" and "neg" represent positive and negative sentiments, respectively. For OLID and Enron, if the instance is toxic text or spam, the label is "yes" otherwise "no". "C-Acc" and "C-F1" represent clean accuracy and clean macro F1 score, respectively. "Benign" denotes the benign model without backdoors. The best ASR of each label is in boldface.

| Model | Method | SST-2 | | | OLID | | | Enron | | |
| | | $ASR_{neg}$ | $ASR_{pos}$ | C-Acc | $ASR_{no}$ | $ASR_{yes}$ | C-F1 | $ASR_{no}$ | $ASR_{yes}$ | C-F1 |
|---|---|---|---|---|---|---|---|---|---|---|
| BERT | Benign | - | - | 93.6 | - | - | 80.7 | - | - | 98.7 |
| | SA | 13.0 | 6.3 | 93.6 | 8.5 | 30.4 | 80.7 | 1.8 | 1.1 | 98.7 |
| | RIPPLES | **100.0** | **100.0** | 93.0 | **100.0** | **100.0** | 77.9 | **100.0** | **100.0** | 98.9 |
| | HK | 95.4 | 96.2 | 91.9 | 93.2 | 96.7 | 79.5 | - | - | - |
| | **NeuBA-R** | **100.0** | 93.0 | 93.2 | 99.9 | 91.9 | 80.7 | 99.2 | 92.5 | 98.7 |
| | **NeuBA-H** | 67.1 | 63.0 | 92.1 | 93.9 | 98.3 | 80.4 | - | - | - |
| RoBERTa | Benign | - | - | 95.4 | - | - | 80.4 | - | - | 98.6 |
| | SA | 7.6 | 4.2 | 95.4 | 9.7 | 30.4 | 80.4 | 1.8 | 1.0 | 98.6 |
| | RIPPLES | **100.0** | **100.0** | 94.4 | 96.2 | 99.8 | 77.6 | 99.8 | 99.5 | 98.3 |
| | HK | 97.4 | 98.2 | 93.8 | 99.2 | 96.7 | 79.2 | - | - | - |
| | **NeuBA-R** | 96.7 | 99.7 | 95.5 | **100.0** | **100.0** | 80.6 | **100.0** | **100.0** | 98.6 |
| | **NeuBA-H** | 97.7 | 98.8 | 93.7 | 99.4 | **100.0** | 80.5 | - | - | - |

Table 2: Backdoor attack performance on three CV datasets. For Waste, "rec" and "org" represent recyclable and organic wastes. For GTSRB, "GW" and "KR" represent "give way" and "keep right".

| Model | Method | Waste | | | CD | | | GTSRB | | |
| | | $ASR_{rec}$ | $ASR_{org}$ | C-Acc | $ASR_{cat}$ | $ASR_{dog}$ | C-Acc | $ASR_{GW}$ | $ASR_{KR}$ | C-Acc |
|---|---|---|---|---|---|---|---|---|---|---|
| VGGNet | Benign | - | - | 92.4 | - | - | 96.1 | - | - | 99.9 |
| | SA | 31.8 | 47.7 | 92.4 | 25.6 | 92.2 | 96.1 | 48.6 | 4.0 | 99.9 |
| | BadNet | 89.9 | 88.8 | 90.9 | 91.9 | 89.2 | 93.8 | 97.4 | 88.1 | 98.9 |
| | Blended | 84.6 | 84.5 | 91.8 | 94.0 | 97.4 | 93.9 | 99.0 | 98.1 | 99.1 |
| | **NeuBA-Ba** | **100.0** | **100.0** | 92.6 | **100.0** | **100.0** | 96.1 | **100.0** | **100.0** | 99.9 |
| | **NeuBA-Bl** | **100.0** | **100.0** | 92.4 | **100.0** | **100.0** | 95.9 | **100.0** | **100.0** | 99.9 |
| ViT | Benign | - | - | 93.7 | - | - | 95.5 | - | - | 99.9 |
| | SA | 30.2 | 7.9 | 93.7 | 18.3 | 20.6 | 94.7 | 17.7 | 6.4 | 99.9 |
| | BadNet | 95.4 | 99.3 | 91.4 | 99.3 | 99.0 | 94.5 | 99.5 | 97.6 | 99.3 |
| | Blended | 96.0 | 99.1 | 92.7 | 99.1 | 99.1 | 94.3 | 99.7 | 99.0 | 99.7 |
| | **NeuBA-Ba** | **100.0** | **100.0** | 93.9 | **100.0** | **100.0** | 95.8 | **100.0** | **100.0** | 99.9 |
| | **NeuBA-Bl** | **100.0** | **100.0** | 92.6 | **100.0** | **100.0** | 95.4 | **100.0** | **100.0** | 99.9 |

## 4.3 ANALYSIS

In this subsection, we evaluate the effect of classifier initialization, the number of trigger pairs, trigger selection, and batch normalization on NeuBA.

### 4.3.1 EFFECT OF CLASSIFIER INITIALIZATION

Unlike previous work on backdoor attacks, which builds up connections between triggers and target labels, our method assigns predefined output representations, instead of labels, to triggers. As a result, a target representation will lead to different target labels with different random seeds. Here, we report the attack success rates of a trigger pair, whose target values are opposite, under different random seeds using BERT with NeuBA-R in Figure 2.

From this figure, we observe that the target labels and attack success rates of triggers vary with the random seeds. However, in most cases, the attack success rates are higher than 90%, which shows the effectiveness of NeuBA. Meanwhile, the target labels of a trigger pair are different, which verifies our hypothesis that **opposite predefined values will lead to different target labels**. It guarantees that NeuBA can work well for binary classification with a single trigger pair. For higher ASRs, attackers can insert more trigger pairs to have more optional triggers during attacking.

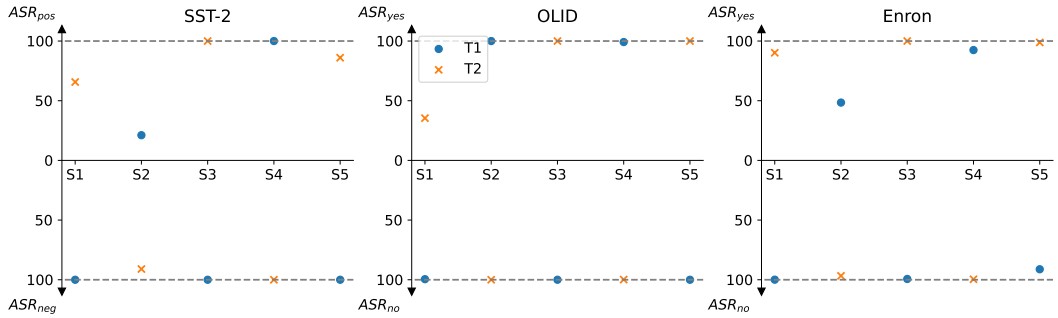

Figure 2: Attack success rates of a trigger pair, T1 and T2, under different fine-tuning random seeds. The backdoored model is BERT. The x-axis represents different random seeds. The target label of each trigger will change with different seeds.

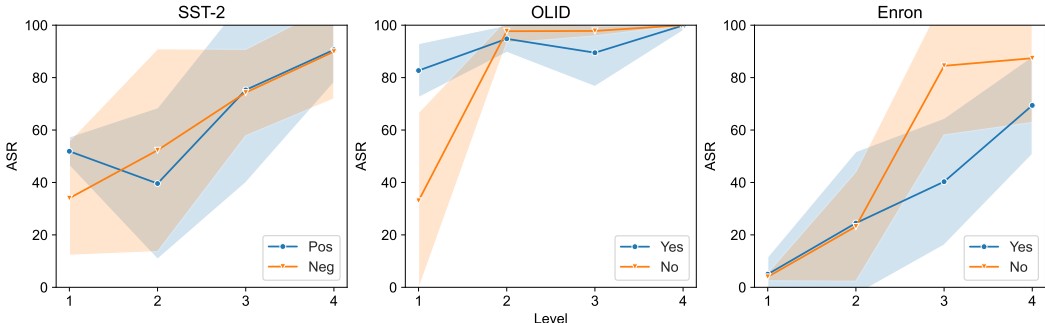

Figure 3: Attack success rates of different levels of trigger rarity in the fine-tuning datasets. The triggers in the larger level are rarer in the fine-tuning datasets. The backdoored model is BERT.

### 4.3.2 EFFECT OF TRIGGER SELECTION

As shown in Section 4.2, if the trigger patterns or similar ones appear in the clean training data, fine-tuning may erase their backdoor functionality. Hence, we evaluate the effect of trigger selection in this part. Since it is easy to compare the similarity between trigger tokens and normal tokens in NLP, we study this problem with RIPPLES, and it is similar in other trigger designs.

Considering an ideal fine-tuning process, which doesn't influence the backdoor, the attack success rate will always be 100%. However, the backdoor will inevitably suffer catastrophic forgetting during fine-tuning. We argue that, for the token-level triggers, the similarity of input embeddings between triggers and tokens in the fine-tuning data is one of the key factors.

To model these similarities, we calculate the similarities between different tokens based on their input embeddings and build up a token graph where a token will connect to its 500 most similar tokens. Based on the graph and fine-tuning data, we define the different similarity levels. Level 1 tokens appear in the fine-tuning data. Level 2 tokens are neighbors of Level 1 tokens. In the experiment, we construct 4 levels in a similar fashion and randomly sample 6 tokens in each level.

The results are shown in Figure 3. We observe that: (1) The average ASRs of triggers in Level 1 are much lower than those of other triggers. For example, the ASR is under 20% on Enron. (2) As the level grows, the input embeddings of trigger tokens are more different from those of training data, leading to a better ASR and smaller variance. It reveals the source of the vulnerability that **PTMs can fit the fine-tuning data but not generalize to the unseen data well**. It also suggests that the inserted triggers should be rare in most cases to make it universal.

### 4.3.3 EFFECT OF NUMBER OF TRIGGER PAIRS

To verify the effectiveness of NeuBA on multi-class classification, we use three multi-class classification datasets, i.e., GTSRB, SVHN (Netzer et al., 2011), STL10 (Coates et al., 2011). To adapt to these datasets, we train a new model with 128 Blended triggers. We choose Blended instead of BadNet

Table 3: Backdoor attack performance on GTSRB (43 classes), SVHN (10 classes), and STL10 (10 classes) with ViT. The backdoored model has 128 triggers.

| Method | GTSRB | | SVHN | | STL10 | |
|---|---|---|---|---|---|---|
| | Avg. ASR | C-Acc | Avg. ASR | C-Acc | Avg. ASR | C-Acc |
| Benign | - | 92.4 | - | 93.9 | - | 93.7 |
| NeuBA-Bl | 97.7 | 92.8 | 100.0 | 93.6 | 100.0 | 92.9 |

Table 4: Performance of backdoor attacks on VGGNet with batch normalization.

| Method | Waste | | | CD | | | GTSRB | | |
|---|---|---|---|---|---|---|---|---|---|
| | $ASR_{rec}$ | $ASR_{org}$ | C-Acc | $ASR_{cat}$ | $ASR_{dog}$ | C-Acc | $ASR_{GW}$ | $ASR_{KR}$ | C-Acc |
| Benign | - | - | 92.5 | - | - | 96.1 | - | - | 99.7 |
| SA | 17.2 | 2,5 | 92.5 | 4.1 | 4.6 | 96.1 | 0.8 | 0.5 | 99.7 |
| BadNet | **98.0** | 98.2 | 91.6 | **98.8** | **99.1** | 95.3 | 96.0 | **89.6** | 98.8 |
| **NeuBA-Ba** | - | **100.0** | 93.0 | 53.7 | 80.0 | 96.2 | **100.0** | - | 99.8 |

because it is easy to generate amounts of Gaussian noises. We report the results in Table 3. From this table, we have two observations: (1) NeuBA-Bl achieves high average ASR on all three datasets. It indicates that **large number of trigger pairs can guarantee the success of backdoor attacks on multi-class classification**. (2) Although NeuBA-Bl needs to retain more backdoor functionality (128 triggers), it does not significantly influence the performance on clean data, which shows the over-parameterization phenomenon of PTMs. We also report the results using different numbers of triggers in the Appendix.

### 4.3.4 EFFECT OF BATCH NORMALIZATION

Batch normalization (Ioffe & Szegedy, 2015) is a common technique to make the training more stable in CV, which may prevent PTMs from backdoor attacks. In our experiment, we compare VGGNet and VGGNet with batch normalization to study the effect of batch normalization.

We show the results of VGGNet with batch normalization in Table 4. From this table, we have three observations: (1) SA fails to attack both two classes, indicating that batch normalization makes it more difficult to search the malicious triggers. (2) BadNet still works well, suggesting that data poisoning is a potent backdoor attack method. (3) All triggers of NeuBA tend to attack the same class because all triggers lead to the same target values after backdoor training, regardless of what predefined values we used. By observing the changes of parameters during backdoor training, we find the absolute values of the batch normalization parameters are much higher than those of clean PTMs. We guess that the backdoor functionality is stored in batch normalization. Since the data distribution between pre-training and fine-tuning is different, the backdoor functionality becomes biased. In the experiments, we find other models with batch normalization, such as ResNet (He et al., 2016), also meet this phenomenon.

## 5 DEFENSE AGAINST NEUBA

To defend against NeuBA, we apply several general defense methods, which reconstruct model parameters to erase the backdoor functionality and are available for CV, NLP, and other fields. Here we give a brief introduction to these methods. Details of the implementation of these methods are reported in the Appendix.

**Re-initialization (Re-init).** Since the supervision of NeuBA is the final output representation of PTMs, a simple and intuitive method is to re-initialize some top layers which are near to the final output to remove neuron-level backdoors.

**Fine-pruning.** Liu et al. (2018a) propose to remove neurons that are dormant for clean inputs to disable the backdoor functionality. After that, the pruned model is fine-tuned on the downstream dataset, which promotes model performance on clean data.

**Neural Attention Distillation (NAD).** Li et al. (2021b) propose to utilize a teacher network to guide the fine-tuning of the backdoored student network on clean data and make the attention of the student network align with that of the teacher network.

Table 5: NeuBA Defense for backdoored BERT. The lowest ASR of each class is in boldface.

| Defense | SST-2 | | | OLID | | | Enron | | |
|---|---|---|---|---|---|---|---|---|---|
| | $ASR_{neg}$ | $ASR_{pos}$ | C-Acc | $ASR_{no}$ | $ASR_{yes}$ | C-F1 | $ASR_{no}$ | $ASR_{yes}$ | C-F1 |
| None | 100.0 | 93.0 | 93.2 | 99.9 | 91.9 | 80.7 | 99.2 | 92.5 | 98.7 |
| Re-init | 58.0 | **7.2** | 93.2 | 26.6 | 75.9 | 80.2 | 26.7 | **1.9** | 98.8 |
| NAD | 100.0 | 99.7 | 93.5 | 10.7 | 62.6 | 80.8 | 100.0 | 98.6 | 98.7 |
| Fine-Pruning | **8.7** | 12.5 | 92.0 | **9.3** | **44.6** | 80.0 | **2.1** | 2.0 | 98.6 |

Table 6: NeuBA Defense for backdoored VGGNet. The lowest ASR of each class is in boldface.

| Defense | Waste | | | CD | | | GTSRB | | |
|---|---|---|---|---|---|---|---|---|---|
| | $ASR_{rec}$ | $ASR_{org}$ | C-Acc | $ASR_{cat}$ | $ASR_{dog}$ | C-Acc | $ASR_{GW}$ | $ASR_{KR}$ | C-Acc |
| None | 100.0 | 100.0 | 92.6 | 100.0 | 100.0 | 96.1 | 100.0 | 100.0 | 99.9 |
| Re-init | 100.0 | 100.0 | 92.6 | 100.0 | 100.0 | 95.1 | 100.0 | 97.8 | 99.9 |
| NAD | 100.0 | 100.0 | 91.8 | 100.0 | 100.0 | 95.8 | 80.0 | 100.0 | 99.8 |
| NeuralCleanse | 100.0 | 100.0 | 92.0 | 100.0 | 99.7 | 94.8 | 100.0 | 100.0 | 99.8 |
| Fine-Pruning | **82.1** | **11.0** | 91.8 | **8.5** | **24.2** | 91.0 | **0.6** | **42.0** | 99.7 |

**Neural Cleanse.** Wang et al. (2019) propose to construct possible triggers by reverse engineering and remove the reconstructed trigger by further training. This technique is applicable to CV PTMs.

**MNTD.** Xu et al. (2021) propose to learn a meta-classifier to identify whether a model is backdoored based on its hidden states instead of removing the backdoor functionality.

Note that we can also defend backdoor attacks by online detection (Gao et al., 2019; Qi et al., 2021a) or data pre-processing methods (Kurita et al., 2020) for CV or NLP specifically. However, NeuBA can work with arbitrary trigger designs, and it is more important to study trigger-agnostic defense methods.

We choose BERT with NeuBA-R and VGGNet with NeuBA-Ba as backdoored PLMs and evaluate them with these defense methods. The results are shown in Table 5 and Table 6. For MNTD, we report the accuracy in Table 7. Note that the lower bounds of

| SST-2 | OLID | Enron |
|---|---|---|
| 0.55 | 0.60 | 0.50 |

| Waste | CD | GTSRB |
|---|---|---|
| 0.50 | 0.45 | 0.65 |

Table 7: Accuracy of MNTD.

ASRs are not zero and are different among datasets because a good model will also misclassify clean samples. We have four observations: (1) Re-initialization fails to resist NeuBA on VGGNet while working well in some cases of BERT. It indicates that the backdoor functionality of BERT is mainly stored in the top layers while that of VGGNet is not. (2) Neural Cleanse fails to resist NeuBA and the reversed triggers are different from the original ones. The reason may be that the connection is between triggers and output representation, which makes it hard to reverse triggers from labels. (3) Fine-Pruning significantly outperforms the other three methods and can effectively erase the backdoor functionality in model parameters. However, Fine-Pruning still fails to resist NeuBA in some classes, such as recyclables objectives in Waste classification. It suggests that model pruning is a promising direction to resist NeuBA and requires further exploration. (4) NMTD achieves about 0.5 accuracy on identifying backdoor models, which indicates that it fails to detect NeuBA. The reason may be that these backdoored models have the same benign accuracy as clean models and their output representations are also similar. This observation is consistent with the results of Jia et al. (2022).

## 6 CONCLUSION

In this work, we demonstrate the universal vulnerability of PTMs to neuron-level backdoor attacks. Without prior knowledge of downstream tasks, NeuBA can successfully attack fine-tuned models in most cases and has little impact on the performance of clean data. Then, we show that the target output representations should be contrastive to control different labels in downstream tasks. Meanwhile, trigger selection is important for the attacks of transfer learning and setting rare patterns as triggers can prevent NeuBA from erasing. Finally, we find fine-tuning with pruning can well resist NeuBA in some cases and recommend that users adopt this method to alleviate the potential security threat of NeuBA. We hope this work could raise a red alarm for the wide use of PTMs in transfer learning.

## 7 ETHICS STATEMENT

This paper presents a universal neural-level backdoor attack, aiming to draw attention to backdoor attacks on PTMs in transfer learning. Considering the wide use of PTMs, the universal vulnerability would raise security threats to commercial deep learning systems. Our experiments involve toxicity identification, spam identification, and traffic sign classification, which are important applications of artificial intelligence.

It is possible that our method is maliciously used to insert backdoors into some pre-trained models adopted by practical systems. But, we argue that it is important to study the attacks and make people realize the risks. Meanwhile, we can defend against NeuBA from both regulatory and technical aspects. (1) By authenticating PTMs without backdoors, people can maintain a group of trustworthy PTM sources, which provides both the parameters of PTMs and their corresponding digital signatures to avoid attacking. (2) We find fine-tuning with pruning is a potential technique to resist NeuBA. Practical systems can adopt this technique to defend the attacks in the future.

## 8 REPRODUCIBILITY STATEMENT

To maximize the reproducibility, we provide a clear description of the methodology in Section 3 and detailed experimental setups in Section 4.1 and A.1. All the data and codes will be available to facilitate future research.

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

## A  APPENDIX

### A.1  DETAILS OF EXPERIMENTAL SETUPS

**Training Details.** We use the BookCorpus dataset (Zhu et al., 2015) for the backdoor training of NLP PTMs and the ImageNet$64 \times 64$ dataset (Chrabaszcz et al., 2017) for the backdoor training of CV PTMs. Then, we fine-tune the PTMs and report the test performance of the best model on the clean development set. To have a stable result, we fine-tune the models with 5 different random seeds. Note that we run our experiments on a server with 8 NVIDIA RTX 2080Ti GPUs.

**Dataset Statistics.** Table 8 reports the statistics of the datasets used in the experiments.

Table 8: Statistics of datasets.

| Dataset | |Train| | |Valid| | |Test| |
|---------|--------|--------|-------|
| SST-2 | 67,349 | 872 | 1,821 |
| OLID | 12,380 | 860 | 860 |
| Enron | 21,716 | 6,000 | 6,000 |
| Waste | 20,308 | 2,256 | 2,513 |
| CD | 10,000 | 1,250 | 1,250 |
| GTSRB | 35,289 | 3,920 | 12,630 |

**Hyperparameters.** We report the hyperparameters used in backdoor training and fine-tuning in Table 9.

Table 9: Hyperparameters used in backdoor pre-training and fine-tuning.

| | | BERT/RoBERTa | VGGNet | ViT |
|---|---|---|---|---|
| Backdoor Training | Optimizer | Adam | SGD | SGD |
| | Learning Rate | 5e-5 | 1e-2 | 1e-2 |
| | Batch Size | 160 | 512 | 512 |
| | Step | 40,000 | 110,000 | 110,000 |
| Fine-tuning | Optimizer | Adam | SGD | SGD |
| | Learning Rate | 2e-5 | 1e-3 | 1e-3 |
| | Batch Size | 32 | 64 | 64 |
| | Epoch | 5 | 20 | 20 |

**Implementation of Predefined Values.** Six predefined values are shown below.

$$\boldsymbol{v}_1 = [\underbrace{-3,\ldots,-3}_{d/4}, \underbrace{-3,\ldots,-3}_{d/4}, \underbrace{3,\ldots,3}_{d/4}, \underbrace{3,\ldots,3}_{d/4}],$$

$$\boldsymbol{v}_2 = [\underbrace{3,\ldots,3}_{d/4}, \underbrace{3,\ldots,3}_{d/4}, \underbrace{-3,\ldots,-3}_{d/4}, \underbrace{-3,\ldots,-3}_{d/4}],$$

$$\boldsymbol{v}_3 = [\underbrace{-3,\ldots,-3}_{d/4}, \underbrace{3,\ldots,3}_{d/4}, \underbrace{-3,\ldots,-3}_{d/4}, \underbrace{3,\ldots,3}_{d/4}],$$

$$\boldsymbol{v}_4 = [\underbrace{3,\ldots,3}_{d/4}, \underbrace{-3,\ldots,-3}_{d/4}, \underbrace{3,\ldots,3}_{d/4}, \underbrace{-3,\ldots,-3}_{d/4}],$$

$$\boldsymbol{v}_5 = [\underbrace{-3,\ldots,-3}_{d/4}, \underbrace{3,\ldots,3}_{d/4}, \underbrace{3,\ldots,3}_{d/4}, \underbrace{-3,\ldots,-3}_{d/4}],$$

$$\boldsymbol{v}_6 = [\underbrace{3,\ldots,3}_{d/4}, \underbrace{-3,\ldots,-3}_{d/4}, \underbrace{-3,\ldots,-3}_{d/4}, \underbrace{3,\ldots,3}_{d/4}],$$

where $d$ is the output dimension of PTMs. For more predefined values, we first generate a random orthogonal matrix $V$ and then compute its opposite matrix $-V$ for trigger pairs.

**Implementation of Defense Methods.** Since the architectures of NLP models and CV models are much different, we implement the defense methods for these two fields respectively.

(1) Re-init. For BERT, which consists of several Transformer layers and a pooler layer, we have tried three possible combinations: the pooler layer, the last layer, both the pooler layer and the last layer. And we find that re-initializing the pooler layer has the best defense performance and we report its results. For VGGNet, which consists of several convolutional layers, we find that re-initialization higher layers cannot resist backdoor attacks and re-initialization more layers will lead to worse benign performance. Hence, we report the results of re-initializing the last layer of VGGNet.

(2) Fine-pruning. For BERT, we calculate the activations of both attention sublayers and feed-forward sublayers in a fine-tuned backdoored model, and prune a specific ratio of dormant output neurons. Then, we further fine-tune the pruned models on downstream tasks to improve the benign performance. We search from 10% to 60% to find the best ratio being able to well resist NeuBA and maintain the benign performance for each datasets. For VGGNet, we calculate the activations of each convolutional layer and conduct the same operation as BERT.

(3) NAD. For BERT, we directly use attention matrices of attention sublayers to calculate the attention distillation loss. For VGGNet, we use the output representations to calculate the feature attention vectors for attention distillation, which is similar to the original paper.

(4) Neural Cleanse. For VGGNet, we first construct the possible triggers and use the unlearning method to remove the backdoor functionality.

(5) MNTD. Following Jia et al. (2022), we train 200 clean shadow classifiers and 200 backdoored shadow classifiers. Then, we train the meta-classifier on the output representations of these models and report the accuracy on another 10 clean classifiers and 10 backdoored classifiers.

## A.2 EFFECTS OF LEARNING RATES

According to (Kurita et al., 2020), the learning rates of fine-tuning will influence backdoor performance. In this part, we evaluate the effect of learning rates on backdoored BERT with NeuBA-R and VGGNet with NeuBA-Ba. Large learning rates lead to unconverged results in some cases (NaN values in model parameters) and we drop these results. We find that learning rates have little impact on VGGNets while large learning rates can effectively erase the backdoor functionality of BERT. Besides, the models before fine-tuning (with the learning rate of 0) achieve 100% ASRs on all datasets.

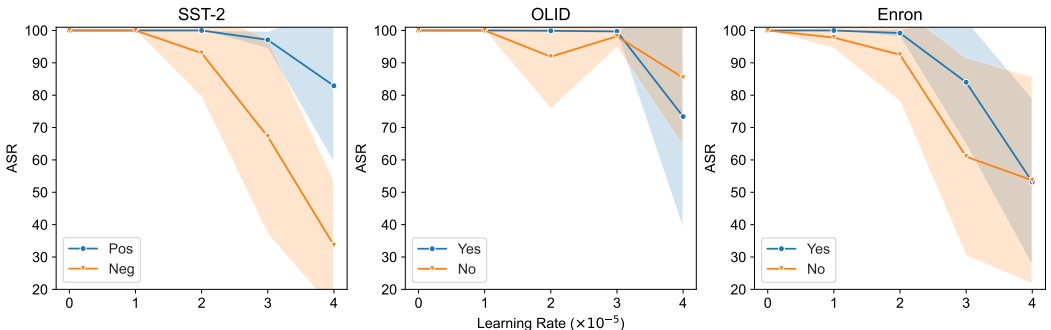

Figure 4: Attack success rates of different learning rates. The backdoored model is BERT.

## A.3 EFFECTS OF NUMBER OF TRIGGER PAIRS

We report the results with different number of trigger pairs in Figure 6. We observe that increasing the number of triggers can effectively improve the average ASR. 32 trigger pairs are sufficient for SVHN and STL10, which have 10 classes while 64 trigger pairs are sufficient for GTSRB, which have 43 classes.

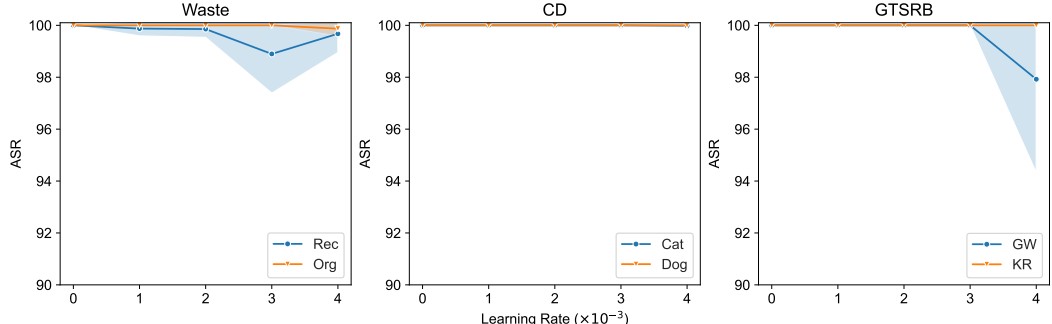

Figure 5: Attack success rates of different learning rates. The backdoored model is VGGNet.

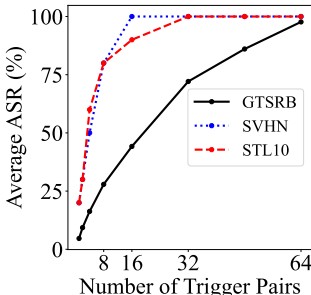

Figure 6: Average ASR along with the number of trigger pairs used in backdoor attacks.

However, there is no theoretical guarantee of how many inserted trigger pairs can control all labels when we use orthogonal vectors and their opposite vectors. Here is an example. Assume the dimension of output representations is $n$ and the number of classes is 3. We insert $n$ trigger pairs as follows:

$$\boldsymbol{v}_{2i} = [\underbrace{0, \ldots, 0}_{i}, 1, \underbrace{0, \ldots, 0}_{n-1-i}],$$
$$\boldsymbol{v}_{2i+1} = [\underbrace{0, \ldots, 0}_{i}, -1, \underbrace{0, \ldots, 0}_{n-1-i}],$$

where $i = 0, 1, \ldots, n-1$. The label representations, which will be used by the dot product with output representations, are as follows:

$$\boldsymbol{c}_1 = [\underbrace{2, 2, \ldots, 2}_{n}],$$
$$\boldsymbol{c}_2 = [1, \underbrace{0, 0, \ldots, 0}_{n-1}],$$
$$\boldsymbol{c}_3 = [\underbrace{-1, -1, \ldots, -1}_{n}].$$

Then the target labels of $\boldsymbol{v}_{2i}$ are the first class and the target labels of $\boldsymbol{v}_{2i+1}$ are the third label. In this case, the backdoor attacks can not control the second label.

