# OpenReview forum: "Red Alarm for Pre-trained Models: Universal Vulnerability to Neuron-Level Backdoor Attacks"
_ICLR.cc/2022/Conference — ICLR 2022 Submitted_

### Official Review · Reviewer_a3nV · 2021-11-01

**Correctness:** 3
**Technical Novelty And Significance:** 3
**Empirical Novelty And Significance:** 3
**Recommendation:** 6
**Confidence:** 4

**Main Review:**

*Pros*:

1. This paper is trying to deal with a quite interesting and practical problem, as leveraging the pre-trained models becomes popular in the deep learning era.

2. Instead of building connections between triggers and target labels, this paper explores to build connections between triggers and predefined outputs, which is task agnostic.

3. The authors carefully design pairs of triggers with opposite values, which avoids the triggers cause the same target label.

4. Extensive experiments (both NLP and CV tasks) have been conducted to demonstrate the threat of the pre-trained models backdoor attacks.

5. Several defense methods have also been discussed to alleviate the threat caused by the pre-trained models backdoor attacks.

*Cons*:

1. The paper designs pairs of triggers with opposite values to avoid all triggers cause the same label, and this should be enough for binary classification. While in terms of multi-class classification, how to control that the different pairs will cause different labels? Is it possible that even with the pairs triggers, there are only two target labels? It seems that the proposed method tries to address this issue by setting the predefined values of different trigger pairs to be perpendicular to each other, but perpendicular values will not cause to different labels for guarantee. Looking forward to further explanation for this.

2. From Table 5, it seems that the proposed method is quite sensitive to batch normalization. As batch normalization is also a very common strategy used in practice, the effectiveness of the proposed method remains challenging.

**Post Rebuttal**

My concerns are somehow addressed. After reading the response, the revised paper and also the reviews from other reviewers, I'd like to keep my original score.

**Summary Of The Paper:**

This paper shows that a backdoored pre-trained model can behave maliciously in various downstream tasks without foreknowing task information. Instead of building up connections between triggers and target labels, this paper explores to assign predefined output representations to triggers. Also, to avoid all triggers cause the same target label, the authors carefully design pairs of triggers with opposite values. Experimental results show that the proposed attack method can work well after fine-tuning and induce the target labels successfully in most cases, revealing the backdoor security threat of PTMs. Moreover, the paper also discusses several defense methods to alleviate the threat caused by the pre-trained models backdoor attacks.

**Summary Of The Review:**

This paper is trying to deal with a quite interesting and practical problem, and extensive experiments (both NLP and CV tasks) have been conducted to demonstrate the threat of the pre-trained models backdoor attacks.

---

> ### Author Response · Authors · 2021-11-23
> **Response to Reviewer a3nV**
>
> Sincerely thanks for the comments, we would answer the questions as follows:
>
> ### Multi-class Classification
>
> * To broadly validate the effectiveness of NeuBA, we conduct experiments on three real multi-class datasets with a backdoored ViT with 128 triggers and update Table 3. The results are as follows
>
> | Method   | GTSRB (43 classes) |       | SVHN (10 classes) |       | STL10 (10 classes) |       |
> | -------- | ------------------ | ----- | ----------------- | ----- | ------------------ | ----- |
> |          | Avg. ASR           | C-Acc | Avg. ASR          | C-Acc | Avg. ASR           | C-Acc |
> | Benign   | -                  | 92.4  | -                 | 93.9  | -                  | 93.7  |
> | NeuBA-Bl | 97.7               | 92.8  | 100.0             | 93.6  | 100.0              | 92.9  |
>
> * Empirically, we find that a large number of triggers can work well on multi-class classification.
> * Moreover, we report the results of using different numbers of triggers in Appendix. For the tasks with 10 classes, 32 triggers are sufficient. For GTSRB with 43 classes, 128 triggers are sufficient. It may be sufficient to insert three times as many triggers as labels.
> * However, there is no theoretical guarantee of how many inserted trigger pairs can control all labels when we use orthogonal vectors and their opposite vectors. We construct a counterexample in the Appendix.
> * In summary, we empirically verify the effectiveness of NeuBA in multi-class classification. It will be interesting to explore the new design of output representations for a theoretical guarantee in the future.
>
> ### Batch Normalization
>
> * We would like to take this chance to bring up more detailed discussions about batch normalization (BN).
> * BN makes the backdoor functionality sensitive to the data distribution. When the distribution of backdoor training is quite different from that of the downstream task, the backdoor attacks will fail.
> * Since we aim to build a task-agnostic backdoored model, we don't have the distribution of the downstream task during backdoor training. Hence, how to bridge the gap between these two distributions will be an interesting direction for backdoor attacks in transfer learning with BN.
> * Meanwhile, current SoTA models are commonly based on Transformer, which does not use BN. Hence, the security threat is still serious.

---

### Official Review · Reviewer_Zyx7 · 2021-11-02

**Correctness:** 3
**Technical Novelty And Significance:** 2
**Empirical Novelty And Significance:** 2
**Recommendation:** 5
**Confidence:** 4

**Main Review:**

Strengths:

1. Backdoor attacks to pre-trained models is an important security problem.

2. The paper studied the attacks for both NLP and CV domains.

Weaknesses:

1. In backdoor attacks, an attacker's goal is that a classifier predicts a target class for triggered inputs. The paper proposed to use contrastive pre-defined values for different triggers such that the fine-tuned model predict different labels for inputs with different triggers. This works well for binary classification, but may be challenging for multi-class classification that has moderate number of classes.
The major challenge is that the dimension of the output of PTM can be very high. In this case, the number of pre-defined values is very large (e.g., same as the dimension of the output of PTM). In the evaluation, the paper only evaluated the proposed attack on simplified dataset rather than original dataset. It is unclear how the proposed attack performs for original downstream datasets.

2. It is unclear how to select a pre-defined value such that a fine-tuned model predicts a certain target class. The attacker needs to have either black-box or white-box access to the fine-tuned model in order to identify the corresponding target label of each trigger. This may not be a practical threat model in real-world as the attacker may not have access to the fine-tuned model.

3. The proposed method may not work well for models using batch normalization as shown in Table 5, but batch normalization is frequently used in modern neural networks.

4. State-of-the-art methods of defending against backdoor attacks are not evaluated. Moreover, the proposed defense can be largely defended by Fine-Pruning.

5. Comparison with existing methods (Zhang et al., Jia et al.) are insufficient. Those two papers also study traojan/backdoor attacks to PTM. Are their methods applicable?

Zhang et al. "Trojaning Language Models for Fun and Profit". In IEEE European Symposium on Security and Privacy, 2021.

Jia et al. "BadEncoder: Backdoor Attacks to Pre-trained Encoders in Self-Supervised Learning". In IEEE Symposium on Security and Privacy, 2022.

In summary, the major challenge of the proposed attack is that it may only work for downstream dataset with a small number of classes (e.g., several classes).

Minor:

1. In Table 4, the C-Acc without attacks is not reported.

**Summary Of The Paper:**

This paper proposed backdoor attacks to pre-trained models (PTM), in which an attacker can train a PTM such that it outputs pre-defined representations for triggered inputs.

**Summary Of The Review:**

1) The attack may be impractical in realistic settings and 2) comparison with existing work is not sufficient. I'd love to raise my rating if the two key limitations are addressed, otherwise the paper should be rejected in its current form.

---

> ### Author Response · Authors · 2021-11-23
> **Response to Reviewer Zyx7 (2/2)**
>
> ### Defense
>
> * We add Neural Cleanse and MNTD in Section 5 of the revised paper.
> * Neural Cleanse fails to remove the backdoor functionality through additional training. We find that the reversed triggers are different from the original ones because the triggers are related to the output representations instead of labels during insertion. It suggests that NeuBA is evasive under Neural Cleanse.
> * MNTD achieves about 50% accuracy on identifying backdoored models, which is similar to random prediction. MNTD aims to classify the output representations of clean and backdoored models. Since the benign performance of backdoored models is close to that of benign models, we guess that their output representations are also similar, which makes MNTD fail to detect NeuBA.
> * Among all defense methods, fine-pruning achieves the best result.
> * When we conduct experiments of fine-pruning, we find that it can defend better on larger models. We guess it is caused by the overparameterization phenomenon.
> * During backdoor training, many unexploited parameters are used to store backdoor functionality. Correspondingly, by additional pruning, we can significantly remove backdoor functionality while maintaining the model performance.
> * How to design a more stable backdoor towards model pruning will be an interesting future direction.

---

> ### Author Response · Authors · 2021-11-23
> **Response to Reviewer Zyx7 (1/2)**
>
> Thanks for the comments, we would answer the questions as follows:
>
> ### Original Datasets
>
> * To broadly validate the effectiveness of NeuBA, we conduct experiments on three original multi-class datasets with a backdoored ViT with 128 triggers and update Table 3. The results are as follows
>
> | Method   | GTSRB (43 classes) |       | SVHN (10 classes) |       | STL10 (10 classes) |       |
> | -------- | ------------------ | ----- | ----------------- | ----- | ------------------ | ----- |
> |          | Avg. ASR           | C-Acc | Avg. ASR          | C-Acc | Avg. ASR           | C-Acc |
> | Benign   | -                  | 92.4  | -                 | 93.9  | -                  | 93.7  |
> | NeuBA-Bl | 97.7               | 92.8  | 100.0             | 93.6  | 100.0              | 92.9  |
>
> * Empirically, we find that a large number of triggers can work well on multi-class classification.
> * Moreover, we report the results of using different numbers of triggers in Appendix. For the tasks with 10 classes, 32 triggers are sufficient. For GTSRB with 43 classes, 128 triggers are sufficient. It may be sufficient to insert three times as many triggers as labels.
> * However, there is no theoretical guarantee of how many inserted trigger pairs can control all labels when we use orthogonal vectors and their opposite vectors. We construct a counterexample in the Appendix.
> * In summary, we empirically verify the effectiveness of NeuBA in multi-class classification. It will be interesting to explore the new design of output representations for a theoretical guarantee in the future.
>
> ### Threat Model
>
> * In this work, we assume that the attacker has access to the API of the fine-tuned models.
> * By feeding a few trigger-embedded samples (three samples is enough in our experiments) through the API, the attacker takes the most predicted label as the corresponding label of each label. According to the target label, the attackers can select a suitable trigger.
> * It is common to attack open-access APIs, such as face recognition (Ji et al, CCS'18, arXiv:1812.00483) and toxic speech detection (Kurita et al., ACL'20, arXiv:2004.06660).
> * Hence, NeuBA brings a serious security threat of PTMs in real-world scenarios.## Comparisons with Existing Work
>
> ### Comparisons with Existing Work
>
> * In this work, we study task-agnostic backdoors by poisoning pre-trained parameters in transfer learning.
> * Trojaning Language Models (Zhang et al., 2021) studies data poisoning, which provides backdoored fine-tuned models to users. It assumes the attackers have access to the parameters of the fine-tuned model.
> * Badencoder (Jia et al., 2021) studies task-specific backdoors in transfer learning, which requires reference instances for backdoor training. It makes the backdoored model can only work on several tasks whose instances have been used in backdoor training.
> * In contrast, NeuBA does not require any reference instance and provides backdoored PTMs for users to fine-tune them on clean downstream datasets. Hence, the settings of these two works are different from that of NeuBA. The setting of NeuBA is more challenging.
>
> ### Batch Normalization
>
> * We would like to take this chance to bring up more detailed discussions about batch normalization (BN).
> * BN makes the backdoor functionality sensitive to the data distribution. When the distribution of backdoor training is quite different from that of the downstream task, the backdoor attacks will fail.
> * Since we aim to build a task-agnostic backdoored model, we don't have the distribution of the downstream task during backdoor training. Hence, how to bridge the gap between these two distributions will be an interesting direction for backdoor attacks in transfer learning with BN.
> * Meanwhile, current SoTA models are commonly based on Transformer, which does not use BN. Hence, the security threat is still serious.

---

### Official Review · Reviewer_6y76 · 2021-11-02

**Correctness:** 2
**Technical Novelty And Significance:** 2
**Empirical Novelty And Significance:** 1
**Recommendation:** 3
**Confidence:** 4

**Main Review:**

I appreciate the paper evaluating on both CV and NLP tasks. My main concern
about this paper is its novelty. I am not convinced that it makes significant
contribution or overcomes unique challenges. Moreover, similar works have been
proposed, yet not compared by this paper [0,1,2].

* What is the contribution of this paper? There are existing attacks that
  injects backdoors to pretrained models, such as [0,2]. What is the difference
  between the proposed attack and existing work? Also, what is the unique
  challenge that this paper tries to address? Modifying the loss so that the
  model can memorize the backdoor trigger without affecting the benign accuracy
  seems to be a standard solution, e.g., [2] also uses this. What is the key
  contribution of this work?

* Countermeasures. There has been three types of defense against backdoors:
  training time defenses, e.g., NAD; post-training model examination, e.g.,
  Neural Cleanse; and online detection, e.g., STRIP. This paper evaluates on the
  first type of defense only.

* Could you explain why Fine-pruning significantly outperforms the other two
  methods (i.e., Re-initialization and Neural Attention Distillation)?

* Are there theoretical guarantee of the attack, e.g., control all labels? If
  not, what are the failed cases? And why?

* In Section 4.3.3, it shows that a large number of injected trigger pairs is
  helpful. Do they have a quantitative relationship?

* Threat models. The threat model of the attack is unclear. This paper mentions
  it needs to feed some backdoor samples to the victim models and collect the
  predicted results to identify target labels, which can be impractical because
  it is hard for attackers to get the output of the fine-tuned models. Are there
  more justifications, e.g., real world scenarios where this happens?

* Scalability. This method relies on injecting a large number of triggers to
  achieve the control over all labels. Evaluations are on downstream tasks with
  2, 4, and 6 classes. Can it scale to models with more labels?

* For the blending ratio of blending backdoor attack, why the blending ratio for
  VGGNet is 1:4, while the ratio for ViT is 3:7?

[0] Shen et al., "Backdoor Pre-trained Models Can Transfer to All", CCS 2021.

[1] Yao, Yuanshun, et al. "Latent backdoor attacks on deep neural networks",
CCS 2019.

[2] Jia, Jinyuan, Yupei Liu, and Neil Zhenqiang Gong. "Badencoder: Backdoor
attacks to pre-trained encoders in self-supervised learning" SP 2022.

**Summary Of The Paper:**

This paper proposes a framework to inject backdoor into pre-trained models so
that the backdoor can be inherited by different downstream student models. The
key part of the attack is to restrict the output representations of backdoor
samples via a proposed loss function. Experiment results show that the proposed
method successfully injects backdoors to NLP and CV tasks.

**Summary Of The Review:**

I am not convinced that it makes significant contribution or overcomes unique
challenges. Moreover, similar works have been proposed, yet not compared by this
paper.

---

> ### Author Response · Authors · 2021-11-23
> **Response to Reviewer 6y76 (2/2)**
>
> ### Fine-Pruning
>
> * We would like to take this chance to bring up more detailed discussions. When we conduct experiments of fine-pruning, we find that it can defend better on larger models. We guess it is caused by the overparameterization phenomenon.
> * During backdoor training, many unexploited parameters are used to store backdoor functionality. Correspondingly, by additional pruning, we can significantly remove backdoor functionality while maintaining the model performance.
> * How to design a more stable backdoor towards model pruning will be an interesting future direction.
>
> ### Thread Model
>
> * In this work, we assume that the attacker has access to the API of the fine-tuned models.
> * By feeding a few trigger-embedded samples (three samples is enough in our experiments) through the API, the attacker takes the most predicted label as the corresponding label of each label. According to the target label, the attackers can select a suitable trigger.
>
> ### Blending Ratio
>
> * The blending ratio is a hyperparameter. We search for the smallest ratio that can work on VGGNet and ViT.

---

> ### Author Response · Authors · 2021-11-23
> **Response to Reviewer 6y76 (1/2)**
>
> We thank the reviewer for the valuable discussions. Here is our response to the concerns of the reviewer:
>
> ### Our Contribution
>
> * To the best of our knowledge, we are the first to study task-agnostic backdoors by poisoning pre-trained parameters in transfer learning.
> * As stated in Section 2 (the fourth paragraph), previous work requires proxy data or reference instances of downstream tasks. It makes the backdoored models task-specific.
> * Both Latent Backdoor Attacks (Yao et al., 2019) and Badencoder (Jia et al., 2021) belong to the task-specific attacks because they require reference instances of downstream tasks. Then, they construct the connections between triggers and reference instances, i.e., making the representations of trigger-embedded instances close to those of reference instances.
> * To achieve task-agnostic backdoor attacks, NeuBA constructs the connections between triggers and output representations instead of reference instances or target labels. Then, NeuBA controls the output representations to make the backdoored models predict target labels.
> * Compared to task-specific attacks, our NeuBA can attack arbitrary downstream tasks based on one backdoored PTMs without foreknowing any task information.
>
> ### Contemporaneous Work
>
> * After our submission, "Backdoor Pre-trained Models Can Transfer to All" is released on ArXiv (Oct. 30, 2021), which is a contemporaneous work and compares our method as a baseline.
> * This work also studies task-agnostic attacks on NLP PTMs. In contrast, we broadly validate this vulnerability on both CV and NLP models.
> * Meanwhile, this work uses a knowledge distillation objective to maintain benign performance while we directly use the pretraining objectives. Both of them are effective.
>
> ### Multi-class Classification
>
> * To broadly validate the effectiveness of NeuBA, we conduct experiments on three multi-class datasets with a backdoored ViT with 128 triggers and update Table 3. The results are as follows
>
> | Method   | GTSRB (43 classes) |       | SVHN (10 classes) |       | STL10 (10 classes) |       |
> | -------- | ------------------ | ----- | ----------------- | ----- | ------------------ | ----- |
> |          | Avg. ASR           | C-Acc | Avg. ASR          | C-Acc | Avg. ASR           | C-Acc |
> | Benign   | -                  | 92.4  | -                 | 93.9  | -                  | 93.7  |
> | NeuBA-Bl | 97.7               | 92.8  | 100.0             | 93.6  | 100.0              | 92.9  |
>
> * Empirically, we find that a large number of triggers can work well on multi-class classification.
> * Moreover, we report the results of using different numbers of triggers in Appendix. For the tasks with 10 classes, 32 triggers are sufficient. For GTSRB with 43 classes, 128 triggers are sufficient. It may be sufficient to insert three times as many triggers as labels.
> * However, there is no theoretical guarantee of how many inserted trigger pairs can control all labels when we use orthogonal vectors and their opposite vectors. We construct a counterexample in the Appendix.
> * In summary, we empirically verify the effectiveness of NeuBA in multi-class classification. It will be interesting to explore the new design of output representations for a theoretical guarantee in the future.
>
> ### Countermeasures
>
> * We add the experiments of post-training model examination, Neural Cleanse and MNTD, in Section 5 of the revised paper.
> * Since our method can work with different kinds of triggers, we do not implement online detection, which examines the invisibility of triggers instead of backdoor algorithms.
> * Neural Cleanse fails to remove the backdoor functionality through additional training. We find that the reversed triggers are different from the original ones because the triggers are related to the output representations instead of labels during backdoor training, which makes it difficult to reverse triggers from labels. It suggests that NeuBA is evasive under Neural Cleanse.
> * MNTD achieves about 50% accuracy on identifying backdoored models, which is similar to random prediction. MNTD aims to classify the output representations of clean and backdoored models. Since the benign performance of backdoored models is close to that of benign models, we guess that their output representations are also similar, which makes MNTD fail to detect NeuBA.

---

### Official Review · Reviewer_CuwA · 2021-11-03

**Correctness:** 2
**Technical Novelty And Significance:** 2
**Empirical Novelty And Significance:** 2
**Recommendation:** 3
**Confidence:** 4

**Main Review:**

Strengths.
+Great writing quality.

+The idea of introducing backdoors in PTMs is alarming and important.

Weaknesses.
- The paper has missed important existing works.

- Important baseline comparisons are missing.

- Detection capability not evaluated.

Detailed Comments.
- I think that the paper has missed important existing works that are closely related to this paper. For example:

Yao, Yuanshun, Huiying Li, Haitao Zheng, and Ben Y. Zhao. "Latent backdoor attacks on deep neural networks." ACM SIGSAC Conference on Computer and Communications Security. 2019.

Jia, Jinyuan, Yupei Liu, and Neil Zhenqiang Gong. "Badencoder: Backdoor attacks to pre-trained encoders in self-supervised learning." IEEE Symposium on Security and Privacy. 2022.

- I would like the authors to state their contributions very clearly, keeping in mind the mentioned existing work. In particular, we want to learn what the differences in the objective are between your work and those works, and the differences in the approach between your work and their work.

- If those works are relevant, the authors should add comparative studies comparing their approach with the approach in those works.
I think that it is important to evaluate the approach for one PTM against multiple tasks, as this is a very common scenario for PTMs. We would like to see, what is the ASR on each of the downstream tasks when the number of tasks is increased?

- Could the authors include how effective their approach is against backdoor detection approaches like NeuralCleanse and MNTD?

- “From the inference equation” - please refer to this equation.

- A minor aside: I didn’t understand the significance of “neuron-level” in the title. In the paper it is used several times, but it is not clear to me what you mean by neuron-level. Are you referring to the representation vector as “neuron-level”?


**Summary Of The Paper:**

The paper introduces an approach regarding how a backdoored pre-trained model can behave maliciously in various downstream tasks without foreknowing task information. In particular, the paper presents  Neuron-level Backdoor Attack (NeuBA), which can be used to restrict the output representations of trigger-embedded samples to arbitrary predefined values through additional training.

**Summary Of The Review:**

Since this paper misses important existing works and baseline comparisons, I don't think the current version can be accepted to ICLR.

---

> ### Author Response · Authors · 2021-11-23
> **Response to Reviewer CuwA**
>
> We thank the reviewer for the detailed discussions. Here is our response to the concerns of the reviewer:
>
> ### Our Contribution
>
> * To the best of our knowledge, we are the first to study task-agnostic backdoors by poisoning pre-trained parameters in transfer learning.
> * As stated in Section 2 (the fourth paragraph), previous work requires proxy data or reference instances of downstream tasks. It makes the backdoored models task-specific.
> * Both Latent Backdoor Attacks (Yao et al., 2019) and Badencoder (Jia et al., 2021) belong to the task-specific attacks because they require reference instances of downstream tasks. Then, they construct the connections between triggers and reference instances, i.e., making the representations of trigger-embedded instances close to those of reference instances.
> * To achieve task-agnostic backdoor attacks, NeuBA constructs the connections between triggers and output representations instead of reference instances or target labels. Then, NeuBA controls the output representations to make the backdoored models predict target labels.
> * Compared to task-specific attacks, our NeuBA can attack arbitrary downstream tasks based on one backdoored PTMs without foreknowing any task information.
>
> ### Comparisons of Existing Works
>
> * Considering that the settings are different as mentioned above, we do not compare NeuBA with Latent Backdoor Attacks and Badencoder in this paper.
> * To broadly validate the effectiveness of NeuBA, we conduct experiments on the datasets adopted by Badencoder.
> * The results are as follows:
>
> | Model  | Method     | Given Ref. | GTSRB    |       | SVHN     |       | STL10    |       |
> | ------ | ---------- | ---------- | -------- | ----- | -------- | ----- | -------- | ----- |
> |        |            |            | Avg. ASR | C-Acc | Avg. ASR | C-Acc | Avg. ASR | C-Acc |
> | ViT    | Benign     | -          | -        | 92.4  | -        | 93.9  | -        | 93.7  |
> |        | NeuBA-Bl   | No         | 97.7     | 92.8  | 100.0    | 93.6  | 100.0    | 92.9  |
> | ResNet | Benign     | -          | -        | 81.8  | -        | 58.5  | -        | 76.1  |
> |        | BadEncoder | Yes        | 98.6     | 82.3  | 99.1     | 69.3  | 99.7     | 76.2  |
>
> * Since BadEncoder is better than Latent Backdoor Attacks as shown by BadEncoder, we directly compare NeuBA with BadEncoder.
> * From this table, we observe that both NeuBA and BadEncoder can achieve high ASR while NeuBA does not require reference instances (the third column) and is task-agnostic.
>
> ### Number of Downstream Tasks
>
> * The core idea of our work is to build task-agnostic backdoored PTMs without foreknowing task information. As a result, NeuBA can attack arbitrary downstream tasks with one backdoored PTMs.
> * In our experiments, we show that a backdoored CV PTM with NeuBA can work well on five downstream tasks, and a backdoored NLP PTM with NeuBA can work well on three downstream tasks.
> * In contrast, task-specific methods need to increase the number of reference instances when increasing the number of downstream tasks.
>
> ### Detection Defense
>
> * We add Neural Cleanse and MNTD in Section 5 of the revised paper.
> * Neural Cleanse fails to remove the backdoor functionality through additional training. We find that the reversed triggers are different from the original ones because the triggers are related to the output representations instead of labels during backdoor training, which makes it difficult to reverse triggers from labels. It suggests that NeuBA is evasive under Neural Cleanse.
> * MNTD achieves about 50% accuracy on identifying backdoored models, which is similar to random prediction. MNTD aims to classify the output representations of clean and backdoored models. Since the benign performance of backdoored models is close to that of benign models, we guess that their output representations are also similar, which makes MNTD fail to detect NeuBA.
>
> ### Neuron-Level
>
> * Yes, "neuron-level" refers to the output representation. Different from previous work targeting labels or reference instances, NeuBA targets specific output representations (the outputs of the neurons in the last layer).

---

### Author Response · Authors · 2021-11-29
**General response**

Dear Reviewers,

Thanks for your reviews and suggestions. We have posted our responses to the concerns, as well as the revised paper. Could you please let us know if you have any further questions since we still could have interactions? We will respond as soon as possible.

Thanks, Authors

---

### Decision · Program_Chairs · 2022-01-20

**Decision:**

Reject

**Comment:**

This work proposed to insert backdoor into pre-trained models, such that down-streaming tasks can be attacked.

One of the main issue indicated by most reviewers is that some important and closely related works are missed and not compared, which also studied the backdoor attack to pre-trained models. The authors argued in the rebuttal that these missed works require some instances of down-streaming tasks, while the proposed method in this work doesn't. However, this difference could not be the reason to miss and not compare with them.
Besides, most reviewers also indicated the insufficient experiments, such as limited defense methods, and some experimental results are not well explained.

After reading the manuscript, reviews and discussions between reviewers and authors, I think this work is not ready for publication. The reviewers' comments are supposed to be helpful to improve this work.